# Housing and Adult Health: Evidence from Chinese General Social Survey (CGSS)

**DOI:** 10.3390/ijerph18030916

**Published:** 2021-01-21

**Authors:** Nanqian Chen, Yanpei Shen, Hailun Liang, Rui Guo

**Affiliations:** 1School of Public Administration and Policy, Renmin University of China, Beijing 100872, China; chennanqian03@163.com (N.C.); syp079@126.com (Y.S.); 2School of Public Health, Capital Medical University, Beijing 100069, China

**Keywords:** housing and health, adult health, Chinese general survey, CGSS

## Abstract

Housing is one of the social determinants of health and the most basic survival needs of human beings. Many studies have preliminarily confirmed that housing factors can influence residents’ health. The aims of this study were: to evaluate the housing factors associated with self-rated health and mental health among Chinese residents; to explore the regional heterogeneity of the impact of housing on health; and to assess the effects of housing on health among different age groups. Data was derived from the Chinese General Social Survey (CGSS). Housing factors were analyzed along six dimensions: housing property, living space, number of living people, number of houses, living region and housing price. Self-rated health and mental health were used to measure health outcomes. Multivariate ordered logistic regression was performed to assess the relationship between housing and health. The living space, living region and housing price was significantly associated with self-rated health. The number of living people living region and housing price were related to mental health. The influence of housing factors on health were more pronounced among residents living in eastern and central area and among the middle-aged group (41–65). Present findings support the notion that housing factors were related to health outcomes. Future studies may focus on the impact of interventions that target on these factors, and the impact of housing on health among special groups such as migrant population and low-and-middle income families.

## 1. Introduction

Access to housing is one of the most fundamental survival needs of human beings. The International Covenant on Economic, Social and Cultural Rights of the United Nations stipulates that everyone should have housing as a fundamental human right. The government of any country or region has the responsibility to ensure that the people it governs enjoy basic housing conditions [1]. Housing is an important social determinant of health. Research showed that improved housing conditions can save lives, prevent disease, improve quality of life, reduce poverty, and help mitigate climate change [1]. It is becoming increasingly important to health in light of urban growth, aging and climate change. With rapid urbanization in China, characterized by radial expansion of urban built-up areas and large-scale migration, the housing problem is more prominent. However, the relationship between housing and health has not been thoroughly investigated in China. As one of the determinants of health, it is of practical and theoretical significance to study the housing and health problems in the context of China. This paper aimed to use Chinese General Social Survey (CGSS) data to explore the relationship between housing conditions and health status, and therefore to conclude implications for policy formulation and adjustment.

Previous studies have preliminarily confirmed that housing factors can influence residents’ health, particularly for vulnerable groups [2,3,4]. People in Ontario (Canada) who lived with HIV and were homeless or unstably housed showed a high prevalence of food insufficiency, and which was associated with physical and mental health-related quality of life (HRQoL) [2]. Each form of housing instability, such as multiple moves and child’s lifetime history of homelessness, was individually associated with increased adjusted odds of adverse health and material hardship, compared with stable housing for caregivers and children among low-income renter households [3]. Inadequate housing and insufficient lighting, unsafe floor surface and furniture created health risks for older people, which would have negative impact on their health [4].

Poor housing quality also leads to health concerns. The majority of physical inspections have identified that housing quality issues, including mold, fire hazard, and others, have the potential to impact human health [5]. Having a high quality, safe and comfortable living environment is the key factor for people to have a high quality and healthy life [6]. Individuals who lived in homes with current cockroach infestations had almost three times the odds of experiencing high depressive symptoms than those without infestation [7]. These studies added important information to the growing body of evidence and showed that housing contributed to population health. Therefore, improvements in population health may not be possible without addressing deficiencies in the housing infrastructure and condition.

Conceptual frameworks have detailed the reasoning underlying the relationship between housing and health [8,9,10]. Housing influences health in three important inter-related aspects, which are the physical conditions within homes, conditions in the neighborhoods surrounding homes and housing affordability [8]. An index of Housing Insults (IHI) captured the means by which housing bundles influence health. IHI had five domains, including affordability, security, quality of dwelling, quality of residential area, and access to services and support [9]. It should be noted that assessing the relationship between health and individual or separate domain of housing is insufficient. Hence, using a broad framework to conceptualize housing is essential. Wan and Su proposed an integrated neighborhood housing deprivation index (INHDI), which was a comprehensive index to combine six dimensions to reflect housing characteristics (internal facilities, living space, physical form and structure, attached facilities, affiliated natural amenities, affordability). They used data at district level of Shenzhen to quantify the association between neighborhood housing deprivation and public health [10].

Previous researches have confirmed that housing has an important impact on health, by using different research methods, perspectives, indicators and research samples. Classical theoretical frameworks are foundation and worthy for current studies to use to screen variables. Exploring the relationship between housing and health in the context of China is the objective of this paper. Housing shortages, rising pricing, residential crowding, and poor housing conditions were common problems in China. However, few studies focused on the influence of housing factors on health, especially on mental health. To bridge the knowledge gap, we used the data from Chinese General Social Survey (CGSS) to examine the relationship between housing and health. The specific aims of this study are: (1) to evaluate whether housing factors were associated with self-rated health and mental health among Chinese residents; (2) to explore the regional heterogeneity of the impact of housing on health; and (3) to explore the effects of housing on health in different age groups.

## 2. Methods

### 2.1. Data

We used data from the 2015 Chinese General Social Survey (CGSS), which is a national, comprehensive and continuous social survey dataset. It was an annual survey started since 2003, which collected data at multiple levels of society, community, family, and individual. We conducted a cross-sectional study. In 2015, respondents were from 478 villages in 28 provinces/cities/autonomous regions in China. The total sample size was 10,968. After dropping 2650 samples with missing value, the final sample size was 8318.

### 2.2. Measures

#### 2.2.1. Dependent Variables

Dependent variables relevant to health outcomes were self-rated health and mental health. Self-rated health and mental health was measured by a single item respectively. Respondents were asked to reply the question “what do you think of your current state of physical health”, and the variable in this study was coded 1–5 as “very unhealthy”, “less healthy”, “generally”, “healthy”, and “very healthy”. They were also asked to reply a question related to mental health “how often have you felt depressed or depressed in the past four weeks”, and this variable was coded 1–5 as “always”, “often”, “sometimes”, “rarely” and “never”. A higher score meant healthier.

#### 2.2.2. Independent Variables

We referred conceptual framework of Emma Baker [9] and extracted variables from CGSS dataset related to housing factors. The main independent variables were housing property [3,11], living space [5,12], number of living people [13], number of houses [14], living region [14] and housing price [15,16,17]. Housing property was measured by a single item, “What type of ownership does your house belong to?” Total answers were categorized into two groups: completely owned and not fully owned (such as owned by parents, owned by spouse’s parents). Living space was coded as a continuous variable. Number of living people was measured by “How many people living together in this house?” (coded as continuous). Respondents were asked to replay question related to number of houses “How many properties do you have?” Answers were also coded as continuous. We generated a dummy variable to describe living region. We divided 28 provinces into three parts, eastern regions (coded as 1), central regions (coded as 2) and western regions (coded as 3). Region not only represented their location differences, but also represented the difference in the level of economic development in China, which is a common regional classification method in China. We also use Real Estate Statistics Database (RESD) to construct a new variable “housing price” to describe average selling price of commercial housing/m^2^ in 2015 in 28 provinces in China (measured in dollars). To reduce collinearity, we take the logarithm of the housing price named “ln housing price” and took “ln housing price” into statistical analysis.

#### 2.2.3. Control Variables

Following the tradition of previous empirical studies, demographic variables and social determinants of health were controlled in all models. Demographic variables included age, gender, marriage, political status [11,15,18]. Social determinants of health variables included income, work status, schooling and hukou (household residence type) [19,20,21,22].

Age was a continuous variable, ranging from 18 to 94. Gender was dichotomized into male (coded as 1) and female (coded as 0). Marriage was dichotomized into having a spouse (coded as1) and not having a spouse (coded as 0). Political status was dichotomized into members of the Communist Party (coded as 1) and other (coded as 0). Income was the personal annual income of respondents (measured in dollars). We generated a dummy variable “ln income” to describe the logarithm of the “income” to reduce collinearity. Work status was dichotomized into employed (coded as 1, including currently engaged in farm work and off-farm work) and not employed (coded as 0). Schooling was a categorical variable assigned from 1 to 5 (under primary school coded as 1, junior high school coded as 2, high school coded as 3, junior college coded as 4 and bachelor degree and above coded as 5). “Hukou” is a household residence type in China. Household registration system is a long-term management practice in China based on urban-rural dual structure, which has a profound impact on urban and rural residents [22]. Hukou was dichotomized into urban (coded as 1) versus rural (coded as 0).

### 2.3. Statistical Analysis

Descriptive analysis of the dependent, independent and control variables was firstly conducted. An ordered logistic regression model was used to examine the relationship between housing and health. We used a conceptual framework to included related variables. Considering both dependent variables could be correlated, therefore we took one as a dependent variable, and controlled the other one in each model. Moreover, we performed two sub-samples ordered logistic regression to assess the impact of housing on health. In the regional sub-sample, we also used ordered logistic regression analysis, separately by each region. Each of them was east (Model 1), central (Model 2) and west (Model 3) in China. Finally, we conducted sub-sample analysis to assess the influence of different age groups on housing and health. Each of them was young adults aging 18 to 30 and 31 to 40 (Model 4 and Model 5), the middle-aged aging 41 to 50 and 51 to 65 (Model 6 and Model 7) and the elderly aging 66 and older (Model 8). All analyses were conducted using STATA software, version 15.0, StataCorp, College Station, TX, USA, Two-tailed *p*-values less than or equal to 0.1 were considered statistically significant. Before statistical analysis, we performed test of correlation coefficient matrix and variance inflation factor (VIF). The absolute values of correlation coefficient ranged from 0.0193 to 0.6147, showing these variables were independent of each other. Furthermore, the mean of VIF value was 1.46, showing that there was no multicollinearity in our model.

## 3. Results

### 3.1. Characteristic of the Sample

Table 1 presents the descriptive statistics for the study variables. The final sample size was 8318. The mean score of self-rated health and mental health were 3.623 and 3.859 respectively, meaning that most of respondents reported that their health status was at general and better levels. 64.3% of respondents reported they had house property. Average living space per person was 48.034 m^2^, and it ranged from 0 to 1200. There were approximately 2 to 3 people living together (including the respondent). The mean value of the number of houses was 1.112, 79% of people reported they had one house. There were 41.1% of respondents lived in Eastern China, 34.7% of them lived in Central China, and 24.20% lived in Western China. The average price of commercial housing sales is 1242.728 dollars/m^2^. The average age of the respondents was 51.344, ranging from 18 to 94. The annual average per capita income was 6054.661dollars. The ratio of male respondents and female respondents is similar. Most are members of the Communist Party. The majority of the respondents had a job (64%). More than half of the respondents had rural hukou. Half of the respondents had elementary and junior high school degrees, the majority of them did not go to college.

### 3.2. Full Sample Analysis

Table 2 presents the ordered logistic regression results. In terms of the independent variables, living space, region and housing price were significantly associated with self-rated health. Living space was positively related to good self-rated health, while region and housing price was negatively related to good self-rated health. The number of living people was positively related to mental health, while the region was negatively related to mental health. Region was a grouping variable, people who lived in Eastern China, Central China and Western China was assigned 1, 2 and 3 respectively. Region was negatively related to self-rated health and mental health, it means that people whose living region were closer to the west, they would like to report bad health status.

In terms of control variables, those who with a higher level of income and education reported good self-rated health and mental health, while age was negatively related to health status. Gender, political status and work status were grouping variables, respondents who were male, members of the Communist Party, reported better self-rated health and mental health. Those who were engaged in work reported good health status.

Considering most variables were not significant, we also carried out a forward stepwise selection regression to obtain a model only including significant variables. In each stepwise regression model, we added terms with *p* < 0.1 and removed those with *p* ≥ 0.2. We also considered one of the two dependent variables and other covariates together. The results were also robust in each model, and details were shown in Appendix A
Table A1 and Appendix A
Table A2.

### 3.3. Sub-Sample Analysis by Region Group

As for the regional heterogeneity of the influence of housing on respondents’ health, this part analyzed the differences of the influence of various housing factors on respondents’ health in eastern, central and western regions.

Table 3 shows the influence of housing factors on self-rated health in living different regions. The sample size of each region was 3417 (eastern), 2889 (central) and 2012 (western), and the mean value of self-rated health scores were 3.713, 3.646 and 3.440 respectively. People living in the eastern region reported higher self-rated health scores than those who lived in the central and western region.

Living space was positively related to self-rated health for all respondents. The impact of living space was also significant to all respondents no matter where they lived (*p* = 0.000, *p* = 0.007 and *p* = 0.045 respectively). Housing price was negatively related to self-rated health for eastern and central residents. However, there was no significance observed in respondents living in the western region. Respondents lived in central region reported good physical health status when they lived with more people.

Table 4 shows the influence of housing factors on mental health in living different regions. Living space was negatively related to mental health to whom lived in the east. The number of living people was positively related to mental health to eastern and western residents. Respondents who lived in central region were more likely reported less mental distress if they owned more houses. Housing price was positively related to mental health among eastern respondents.

### 3.4. Sub-Sample Analysis by Age Group

We also divided the sample into five groups by age, group 1 and group 2 were young adults aging 18 to 30 and 31 to 40 respectively, group 3 and group 4 was middle-aged people aging 41 to 50 and 51 to 65 respectively, group 5 was the elderly aging 66 and older.

Table 5 and Table 6 presents the results of the influence of housing factors on health status in different age groups. People’s self-rated health scores declined with age. Whatever age group were respondents in, region and housing price had significant negative effects on self-rated health. The middle-aged (51–65) and the elderly (66 and over) respondents were more likely reported good self-rated health if they lived in lager houses. The impact of living space was more significantly to the middle-aged. But we cannot observe significance for young adults (18–40) on the variable of living houses.

In terms of mental health (Table 6), the results look a little different. Respondents who were aged between 31–50 reported better mental health status if they completely owned this house. The number of living people positively related to mental health for those who were 18 to 30 years old and 51 to 65 years old. Region was negatively related to all age group respondents except the elderly. Housing price was significantly associated with mental health for people aged over 41.

## 4. Discussion

This study examined the relationship between housing conditions and health status among a representative sample of 8318 residents in China. To our knowledge, there are few studies that assess the associations between housing factors and health in the Chinese population. Our study provides evidence of the associations of health with several housing factors, living space, number of living people, region and housing price. When the sample was stratified by geographical areas and age groups, the results were persistent and robust. However, in different groups, some housing factors showed stronger effects, while some showed weaker influence. In addition, similar to previous researches showed, some traditional social economic factors significantly influenced health, such as age, income, gender education.

Significant differences were observed in health outcomes across various housing characteristics. Living space was positively related to self-rated health, while living region was negatively related to health. People living in a crowded house were more likely to get infectious diseases (including TB, gastroenteritis and diarrheal diseases) [12,23]. Also, less crowding was conducive to better parental housing conditions [24]. Several studies found that crowding was linked to psychological distress and behavior problems among children [24,25,26]. However, we did not observe significance between living space and mental health among adults in this study. WHO (2018)’ report also indicated that the evidence relating to adverse mental health effects was assessed as moderated to low [12].

We found that number of people who live in a house was positively related to mental health. This result is also robust among respondents aging 66 and over. The empty nester and the elderly living alone are expanding in China. Previous studies have shown that living with children is an important dimension for the elderly to obtain social support. Living together allowed the elderly to gain more emotional support or instrumental support, thereby reducing depression and improving the happiness of the elderly [13,27].

Higher housing price was negatively related to bad self-rated health while we observed a significant positive influence of housing price on mental health. In recent years, the sales price of domestic commercial housing has been rising, burdening people a lot. Rising housing prices lead to the decline of family housing affordability, housing price as living cost deprived them of their income, and their health will be affected [17]. In order to buy a house and pay the rent, they have to reduce living expenses related to healthy food, medical care and health services [25,28]. Renters who reported living in unaffordable housing showed the likelihood of having depressive symptoms and poor self-rated health [11]. This study also found that housing price was positively related to reduced mental distress, which was in line with previous studies [29]. Housing prices link to area characteristics. Higher housing prices reflected better regional amenities and economic opportunities, which will reduce people’s uncertainty about the future economic development and increase their expected income. Therefore, housing price could have a direct impact on mental health [29].

Our study found that the influence of housing on health had regional heterogeneity. We can observe a more significant influence on those who lived in Eastern and Central China, while the impact of housing on health is less pronounced in Western China. Living space was positively related to health while housing price negatively related to health. Due to the unbalanced regional economic development of China, the unbalanced distribution of public service resources in different regions is very prominent [15]. The eastern and central regions have a higher level of economic development and the process of urbanization has entered a very fast developing period. Residents lived in these areas having higher income than those lived in the west area. Therefore, they could afford larger houses. Housing quality improvements lead to diminished mental distress [26,30]. At the same time, because of the relatively high housing prices in economically developed areas, the marginal effect of housing prices on health is more obvious.

The influence of housing factors on health varied among different age groups, and were more pronounced in the middle-aged group (aged 41–65). The middle-aged group paid more attention to housing property, living space, housing price, region. Residents of different age groups have different evaluations on housing impact, for example, although younger people are aware of the importance of housing, they are in wealth accumulation stage and cannot buy their own house in early years, which makes the housing effects on health of the young is not so significant [14]. However, the middle-aged group bears the dual responsibility of raising children and providing care for their aged parents. Hence the demand for housing factors is greater. With the tradition of an obsession with housing ownership housing is more than a physical residence, it can provide a great sense of prestige and emotional stability as well [25].

This study confirmed the relationship between housing and health in the context of China from an empirical perspective, and also linked the Real Estate Statistics Database (RESD) with the inclusion of housing price as a variable. In addition, we provided two sub-sample studies (regional grouping and age grouping), which provided new ideas for its localization research in China. Moreover, this study proved that there is little relationship between owning housing property and residents’ health, which provides a data basis for the further promotion of “equal right to rent and purchase” policy.

However, there are some limitations in our studies that should be mentioned. Due to data limitations, the studied housing variables are relatively few, and there is a lack of variables to measure housing quality and neighborhood environment in a micro way. Second, many studies have shown that housing plays an important role in children’s growth and health, but there is currently no data on children under the age of 18 in CGSS. Third, in the sub-sample analysis, the coefficients of every model varied. However, it is difficult to compare the coefficients difference in three or more separated models in sub-ample analysis at the same time or examined the influence of interaction items on key variables in the same model.

With the reform on land use and housing system, the housing price level had a long-term rising trend since 1999, exorbitant house prices cannot ensure that everyone has decent housing. Firstly, the government should further strengthen the housing fund and other security systems, improving the real estate market. Secondly, housing policy should focus on regional differences, and pay attention to urban housing problems in eastern and central regions, especially to solve the problems of high housing prices and crowded per capita living space. Thirdly, advancing some housing assistance programs is important, such as public housing, and multifamily housing programs [31].

## 5. Conclusions

We used data from the 2015 Chinese General Social Survey (CGSS) to examine the associations between housing factors and health status. The findings suggest that living space, the number of living people, region and housing price are significantly related to health. We also found the influence of housing on health with regional heterogeneity. The impact is greater to respondents living the eastern and central regions. Housing impact on health also varied in different age groups. Middle-aged people were more sensitive to the effects of housing on health. From a practical perspective, this data can be used to further study the housing problems of special groups of migrant population, to explore the relationship between housing and health through housing stability, and to investigate the similar topics among low-and-middle-income families.

## Figures and Tables

**Table 1 ijerph-18-00916-t001:** Descriptive statistics of respondents’ demographic variables and housing factors.

Variables	Observations	Mean	Standard Deviation	Min	Max
Self-rated health	8318	3.623	1.051	1	5
Mental health	8318	3.859	0.910	1	5
Housing property	8318	0.643	0.479	0	1
Living space	8318	48.034	44.205	0	1200
Number of living people	8318	2.813	1.468	1	50
Number of houses	8318	1.112	0.604	0	14
Region	8318	1.831	0.790	1	3
Housing price	8318	1242.728	823.4091	691.028	3633.839
ln housing price	8318	6.987	0.468	6.540	8.198
Age	8318	51.344	16.191	18	94
Income	8318	6054.661	33,155.32	8.027	1,605,547
ln income	8318	7.921	1.268	2.083	14.289
Marriage	8318	0.8045	0.397	0	1
Gender	8318	0.512	0.500	0	1
Politic identity	8318	0.120	0.325	0	1
Hukou	8318	0.461	0.499	0	1
Work status	8318	0.640	0.480	0	1
Schooling	8318	2.234	1.254	1	5

“Hukou” refers to a household residence type in China. Household registration system is a long-term management practice in China based on urban-rural dual structure. The hukou attribute has a profound impact on urban and rural residents. “Income” refers to annual personal income. To reduce collinearity, we took logarithm of “income” and generated a new variable called “ln income”. Then we took “ln income” into analysis. “Housing price” was average selling price of commercial housing/m^2^ in 2015 in China. To reduce collinearity, we took logarithm of “housing price” and generated a new variable called “ln housing price”. Then we took “ln housing price” into analysis.

**Table 2 ijerph-18-00916-t002:** Results for full sample ordered logistic regression.

	Self-Rated Health		Mental Health	
Variables	Coefficient	*p*-Value	Coefficient	*p*-Value
Housing property	−0.043 (0.046)	0.356	0.029 (0.047)	0.531
Living space	0.002 (0.001)	0.000 ***	−0.000 (0.001)	0.735
Number of living people	0.026 (0.016)	0.114	0.052 (0.017)	0.003 ***
Number of houses	−0.011 (0.035)	0.743	0.027 (0.036)	0.451
Region	−0.216 (0.034)	0.000 ***	−0.179 (0.034)	0.000 ***
ln housing price	−0.611 (0.059)	0.000 ***	0.299 (0.062)	0.000 ***
Mental health	0.932 (0.026)	0.000 ***		
Self-rated health			0.864 (0.024)	0.000 ***
Age	−0.035 (0.002)	0.000 ***	0.008 (0.002)	0.000 ***
ln income	0.167 (0.021)	0.000 ***	0.080 (0.021)	0.000 ***
Marriage	−0.068 (0.056)	0.226	0.104 (0.057)	0.068 *
Gender	0.103 (0.043)	0.017 **	0.072 (0.043)	0.097 *
Politic identity	0.127 (0.070)	0.069 *	0.108 (0.071)	0.125
Hukou	−0.032 (0.054)	0.559	0.071 (0.054)	0.192
Work status	0.277 (0.052)	0.000 ***	−0.124 (0.052)	0.017 **
Schooling	0.061 (0.023)	0.008 ***	0.052 (0.023)	0.025 **
LR chi^2^ (15)	3056.34		1993.04	
Prob > chi^2^	0.0000 ***		0.0000 ***	
N	8318		8318	

“Hukou” refers to a household residence type in China. Household registration system is a long-term management practice in China based on urban-rural dual structure. The hukou attribute has a profound impact on urban and rural residents. “Income” refers to annual personal income. To reduce collinearity, we took logarithm of “income” and generated a new variable called “ln income”. Then we took “ln income” into analysis. “Housing price” was average selling price of commercial housing/m^2^ in 2015 in China. To reduce collinearity, we took logarithm of “housing price” and generated a new variable called “ln housing price”. Then we took “ln housing price” into analysis. * *p* < 0.1; ** *p* < 0.05; *** *p* < 0.01 (two sided; standard errors in parentheses).

**Table 3 ijerph-18-00916-t003:** Results for the impact of housing on self-rated health by region group.

	Model 1		Model 2		Model 3	
	East		Central		West	
Variables	Coefficient	*p*-Value	Coefficient	*p*-Value	Coefficient	*p*-Value
Housing property	0.007 (0.072)	0.927	−0.014 (0.081)	0.866	-0.126 (0.095)	0.185
Living space	0.004 (0.001)	0.000 ***	0.002 (0.001)	0.007 ***	0.002 (0.001)	0.045 **
Number of living people	0.004 (0.024)	0.861	0.063 (0.030)	0.036 **	0.015 (0.034)	0.669
Number of houses	−0.041 (0.049)	0.402	−0.050 (0.076)	0.511	0.088 (0.075)	0.240
ln housing price	−0.573 (0.073)	0.000 ***	−0.707 (0.355)	0.046 **	−0.637 (0.559)	0.255
Mental health	0.805 (0.039)	0.000 ***	1.023 (0.045)	0.000 ***	1.048 (0.054)	0.000 ***
Age	−0.039 (0.003)	0.000 ***	−0.032 (0.003)	0.000 ***	−0.034 (0.004)	0.000 ***
ln income	0.208 (0.038)	0.000 ***	0.099 (0.035)	0.005 ***	0.192 (0.040)	0.000 ***
Marriage	0.065 (0.085)	0.445	−0.260 (0.102)	0.010 ***	0.017 (0.114)	0.879
Gender	0.186 (0.067)	0.006 ***	−0.040 (0.074)	0.593	0.136 (0.087)	0.116
Politic identity	0.058 (0.096)	0.543	0.359 (0.128)	0.005 ***	0.041 (0.170)	0.810
Hukou	−0.008 (0.088)	0.925	−0.113 (0.090)	0.207	0.005 (0.113)	0.965
Work status	0.320 (0.092)	0.001 ***	0.381 (0.084)	0.000 ***	0.109 (0.101)	0.280
Schooling	−0.018 (0.034)	0.588	0.138 (0.042)	0.001 ***	0.108 (0.052)	0.039 **
LR chi^2^ (14)	1191.60		1092.01		767.66	
Prob > chi^2^	0.0000 ***		0.0000 ***		0.0000 ***	
N	3417		2889		2012	

“Hukou” refers to a household residence type in China. Household registration system is a long-term management practice in China based on urban-rural dual structure. The hukou attribute has a profound impact on urban and rural residents. “Income” refers to annual personal income. To reduce collinearity, we took logarithm of “income” and generated a new variable called “ln income”. Then we took “ln income” into analysis. “Housing price” was average selling price of commercial housing/m^2^ in 2015 in China. To reduce collinearity, we took logarithm of “housing price” and generated a new variable called “ln housing price”. Then we took “ln housing price” into analysis. ** *p* < 0.05; *** *p* < 0.01 (two sided; standard errors in parentheses).

**Table 4 ijerph-18-00916-t004:** Results for the impact of housing on mental health by region group.

	Model 1		Model 2		Model 3	
	East		Central		West	
Variables	Coefficient	*p*-Value	Coefficient	*p*-Value	Coefficient	*p*-Value
Housing property	0.026 (0.073)	0.724	0.036 (0.081)	0.661	0.031 (0.096)	0.747
Living space	−0.003 (0.001)	0.003 ***	0.001 (0.001)	0.347	0.001 (0.001)	0.283
Number of living people	0.046 (0.027)	0.086 *	0.039 (0.030)	0.181	0.070 (0.034)	0.040 **
Number of houses	0.030 (0.049)	0.542	0.153 (0.078)	0.050 **	−0.085 (0.069)	0.218
ln housing price	0.332 (0.075)	0.000 ***	−0.040 (0.361)	0.912	−0.457 (0.563)	0.417
Self-rated health	0.787 (0.039)	0.000 ***	0.910 (0.040)	0.000 ***	0.945 (0.048)	0.000 ***
Age	0.006 (0.003)	0.034 **	0.007 (0.003)	0.018 **	0.013 (0.004)	0.001 ***
ln income	0.038 (0.037)	0.300	0.139 *** (0.035)	0.000	0.038 (0.040)	0.337
Marriage	0.030 (0.086)	0.722	0.180 (0.102)	0.078 *	0.071 (0.116)	0.541
Gender	0.045 (0.067)	0.501	0.122 (0.075)	0.105	0.051 (0.088)	0.559
Politic identity	0.208 (0.098)	0.033 **	−0.004 (0.129)	0.976	0.039 (0.171)	0.821
Hukou	0.132 (0.087)	0.129	−0.077 (0.091)	0.398	0.202 (0.114)	0.076 *
Work status	−0.126 (0.092)	0.171	−0.121 (0.085)	0.154	−0.111 (0.103)	0.278
Schooling	0.033 (0.034)	0.328	0.059 (0.042)	0.160	0.085 (0.053)	0.109
LR chi^2^ (14)	543.70		719.75		513.58	
Prob > chi^2^	0.0000 ***		0.0000 ***		0.0000 ***	
N	3417		2889		2012	

“Hukou” refers to a household residence type in China. Household registration system is a long-term management practice in China based on urban-rural dual structure. The hukou attribute has a profound impact on urban and rural residents. “Income” refers to annual personal income. To reduce collinearity, we took logarithm of “income” and generated a new variable called “ln income”. Then we took “ln income” into analysis. “Housing price” was average selling price of commercial housing/m^2^ in 2015 in China. To reduce collinearity, we took logarithm of “housing price” and generated a new variable called “ln housing price”. Then we took “ln housing price” into analysis. * *p* < 0.1; ** *p* < 0.05; *** *p* < 0.01 (two sided; standard errors in parentheses).

**Table 5 ijerph-18-00916-t005:** Results for the impact of housing on self-rated health by age group.

	Model 4	Model 5	Mode l6	Model 7	Model 8
	18–30	31–40	41–50	51–65	66–94
Variables	Coefficient	Coefficient	Coefficient	Coefficient	Coefficient
Housing property	0.119 (0.158)	−0.136 (0.119)	0.077 (0.111)	−0.047 (0.090)	0.073 (0.097)
Living space	0.003 (0.002)	0.005 ** (0.002)	0.002 (0.001)	0.003 *** (0.001)	0.003 ** (0.001)
Number of living people	−0.034 (0.046)	0.034 (0.033)	0.044 (0.042)	0.004 (0.031)	0.065 (0.042)
Number of houses	−0.058 (0.082)	−0.049 (0.098)	0.030 (0.084)	−0.002 (0.066)	−0.066 (0.081)
Region	−0.241 ** (0.098)	−0.112 (0.091)	−0.237 *** (0.072)	−0.178 *** (0.061)	−0.254 *** (0.075)
ln house price	−0.287 * (0.170)	−0.451 *** (0.155)	−0.882 *** (0.147)	−0.594 *** (0.108)	−0.620 *** (0.123)
Mental health	0.618 *** (0.075)	0.739 *** (0.071)	1.065 *** (0.057)	0.948 *** (0.047)	1.051 *** (0.055)
Age	−0.060 *** (0.022)	−0.033 (0.020)	-0.068 *** (0.016)	−0.029 *** (0.009)	−0.009 (0.008)
ln income	−0.022 (0.067)	0.236 *** (0.066)	0.315 *** (0.050)	0.194 *** (0.041)	0.116 ** (0.047)
Marriage	0.169 (0.153)	0.275 (0.198)	0.238 (0.172)	0.062 (0.114)	−0.140 (0.108)
Gender	0.227 * (0.124)	0.076 (0.116)	-0.058 (0.094)	0.053 (0.080)	0.241** (0.097)
Politic identity	0.562 ** (0.241)	0.212 (0.199)	-0.012 (0.183)	-0.041 (0.124)	0.158 (0.125)
Hukou	−0.096 (0.142)	−0.052 (0.142)	0.006 (0.117)	0.068 (0.105)	0.034 (0.136)
Work status	0.011 (0.181)	0.357 * (0.202)	0.164 (0.131)	0.464 *** (0.089)	0.381 *** (0.122)
Schooling	0.094 (0.058)	0.004 (0.061)	0.040 (0.054)	0.081 * (0.049)	−0.035 (0.050)
LR chi^2^ (15)	112.70	172.62	576.98	637.75	493.03
Prob > chi^2^	0.0000	0.0000	0.0000	0.0000	0.0000
N	1060	1196	1804	2506	1752

“Hukou” refers to a household residence type in China. Household registration system is a long-term management practice in China based on urban-rural dual structure. The hukou attribute has a profound impact on urban and rural residents. “Income” refers to annual personal income. To reduce collinearity, we took logarithm of “income” and generated a new variable called “ln income”. Then we took “ln income” into analysis. “Housing price” was average selling price of commercial housing/m^2^ in 2015 in China. To reduce collinearity, we took logarithm of “housing price” and generated a new variable called “ln housing price”. Then we took “ln housing price” into analysis. * *p* < 0.1; ** *p* < 0.05; *** *p* < 0.01 (two sided; standard errors in parentheses).

**Table 6 ijerph-18-00916-t006:** Results for the impact of housing on mental health by age group.

	Model 4	Model 5	Model 6	Model 7	Model 8
	18–30	31–40	41–50	51–65	66–94
Variables	Coefficient	Coefficient	Coefficient	Coefficient	Coefficient
Housing property	0.100 (0.154)	0.211 * (0.118)	0.300 *** (0.112)	−0.158* (0.092)	−0.009 (0.099)
Living space	0.000 (0.002)	−0.003 (0.002)	−0.001 (0.001)	0.001 (0.001)	−0.002 (0.001)
Number of living people	0.149 ** (0.045)	0.006 (0.032)	−0.033 (0.041)	0.068 ** (0.031)	0.039 (0.042)
Number of houses	−0.145 ** (0.073)	0.040 (0.097)	0.037 (0.084)	0.054 (0.071)	0.100 (0.082)
Region	−0.223 ** (0.094)	−0.363 *** (0.090)	−0.159 ** (0.073)	−0.144 ** (0.062)	−0.085 (0.076)
ln house price	0.024 (0.171)	−0.132 (0.155)	0.600 *** (0.155)	0.404 *** (0.114)	0.421 *** (0.131)
Self-rated health	0.701 *** (0.079)	0.717 *** (0.069)	1.009 *** (0.054)	0.837 *** (0.042)	0.917 *** (0.049)
Age	−0.039 * (0.022)	0.026 (0.020)	0.006 (0.016)	0.030 *** (0.001)	−0.010 (0.008)
ln income	0.068 (0.063)	0.098 (0.063)	0.052 (0.050)	0.111 *** (0.041)	0.047 (0.048)
Marriage	0.174 (0.149)	0.080 (0.196)	0.290 * (0.176)	0.392 *** (0.118)	−0.168 (0.110)
Gender	−0.012 (0.120)	0.148 (0.115)	0.1408 (0.095)	0.120 (0.082)	0.005 (0.099)
Politic identity	0.122 (0.232)	−0.115 (0.197)	−0.027 (0.185)	0.086 (0.128)	0.199 (0.130)
Hukou	0.034 (0.138)	0.002 (0.141)	0.080 (0.117)	0.122 (0.107)	0.080 (0.139)
Work status	0.338 ** (0.175)	0.336 * (0.199)	−0.023 (0.132)	−0.203 ** (0.090)	−0.072 (0.123)
schooling	0.100 * (0.056)	−0.023 (0.061)	0.039 (0.054)	0.096 ** (0.050)	0.100 * (0.053)
LR chi^2^ (15)	127.97	177.88	541.10	672.82	496.48
Prob > chi^2^	0.0000	0.0000	0.0000	0.0000	0.0000
N	1060	1196	1804	2506	1752

“Hukou” refers to a household residence type in China. Household registration system is a long-term management practice in China based on urban-rural dual structure. The hukou attribute has a profound impact on urban and rural residents. “Income” refers to annual personal income. To reduce collinearity, we took logarithm of “income” and generated a new variable called “ln income”. Then we took “ln income” into analysis. “Housing price” was average selling price of commercial housing/m^2^ in 2015 in China. To reduce collinearity, we took logarithm of “housing price” and generated a new variable called “ln housing price”. Then we took “ln housing price” into analysis. * *p* < 0.1; ** *p* < 0.05; *** *p* < 0.01 (two sided; standard errors in parentheses).

## Data Availability

Publicly available datasets were analyzed in this study. This data can be found here: http://www.cnsda.org/index.php?r=projects/view&id=62072446.

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
