# Peer review of "Housing and Adult Health: Evidence from Chinese General Social Survey (CGSS)"

_ijerph, 2021, doi:10.3390/ijerph18030916_

Round 1

Reviewer 1 Report

Many thanks to the authors for spending more time in rectifying the issues I noticed in the first place. Overall, the response and the changes made are looking good to me. I hope the authors agree along with me that the changes made the paper even more interesting and in good shape. 

Few point-to-point observations:

  1. Happy with the response for point 1.
  2. Generally happy with how the section is re-written. There are some grammatical inconsistencies, I hope the author or the copy editing team can rectify them. Again naming of the models could be in a continuous form, for example, Model 1-3 for regions and model 4-8 for age groups. This would remove some confusions.
  3. Generally happy with point 3.
  4. Though the authors have indicated they made the issue of lower R-squared values in the limitations section, I couldn’t see it explicitly. Was it missed out by accident?

Best wishes

R

Reviewer 2 Report

Here are some suggestions for the authors:

Results in Table 2 give details on what variables are statistically significant. However, I would like to know the statistically significance of the whole model. Another interesting topic on this table is the difference between the two models. We can see that some variables are significant in one model, but are not significant in the other model. Why? Even more, the sign of some coefficients change from one model to the other. A deeper explanation on this point could help to understand these results.

Table 3 gives results for the impact of housing on self-rated health by region group. We can see differences in coefficients, but we need to know whether these differences are statistically significant. For example, we can observe that age gets the coefficients, -0.039, -0.032 and -0.034. But maybe these differences are not statistically significant, so we could conclude that there is no difference in the age importance regardless on the region. Similar analysis for Tables 4-6. This is a very important point that should be address by authors. There are well known statistical techniques to face this point.

Authors could include some limitations of the research.

Reviewer 3 Report

Comments and Suggestions

Notwithstanding the use of mental health measurement scales, the authors did use secondary data “due to unavailability of measurement of EQ-5D, SF-36 and other indicators” (CGSS). 

That is why I have not commented on the use of data from a derived source. On the other hand, the notion of "area" was well evidenced by the authors in the initial version.

It is necessary to realize the heterogeneity of the Chinese reality with regard to housing, which varies greatly from region to region. This is a strong way to understand the effects of housing on mental health, summarized in Table 6.

The authors used multivariate regression to try to correlate housing with health. The authors also have not forgotten the direct connection between housing and disposable income per household, whose correlation exists in the short and long term (eg, Alessio Fusco - "The relationship between income and housing deprivation: A Longitudinal Analysis", Economic Modeling, Volume 49, September 2015, Pages 137-143, and Bentley, Rebecca; Baker, Emma; Aitken, Zoe, “The 'double precarity' of employment insecurity and unaffordable housing and its impact on mental health”, Social Science & Medicine. Mar 2019, Vol. 225, p 9-16).

From the explanation given by the authors, I suggest that the concept of “political status” be maintained instead of “political identity”.

Round 2

Reviewer 2 Report

Authors have addressed the concerned points.

This manuscript is a resubmission of an earlier submission. The following is a list of the peer review reports and author responses from that submission.

Round 1

Reviewer 1 Report

Detailed comments:

1. We usually use scales to measure the degree of mental health. Meanwhile, there are also measures of physical health such as EQ-5D, SF-36, etc. Therefore, the authors would be wise to explain why self-rated health and whether felt depressed were chosen as proxy variables of health. Are there any other options in the questionnaire?

2. As far as I know, living area is a categorical variable. However, why do the living area variable in the regression results show the form of continuous variables?

3. In table 2, what makes me wonder is that the Coefficient of age square is 0.000, which means that this variable has little effect on the dependent variable, but why is there such a strong statistical significance?

4. In the method section, Schooling was a continuous variable assigned from 1 to 13. However, Table 1 in result section showed Schooling ranges from 1 to 14. What is the cause of the inconsistency?

Reviewer 2 Report

The article, “Housing and Adult Health: Evidence from Chinese General Social Survey (CGSS)” seeks to address the existing gap in knowledge about the relationship between housing conditions and health status. The issues raised in the paper are of significance in exploring and addressing the health gaps with special references to highly urbanising places where housing prices is particularly (and unaffordable) high. The authors have effectively used the 2015 round of Chinese General Social Survey to assess the health gap by regions and the explanation of this gap with reference to the housing conditions.

The topic itself is of great importance, especially because it is clearly highlighted that housing is one of the basic human needs. There are several notable aspects in the manuscript: the hierarchical nature of the data and inclusion of secondary datasets or computation of indices. Moreover, exploration of the role of the contextual factors by controlling the role of individual factors is of great significance. Having professed my general enthusiasm for the topic and its importance, I have some concerns that I feel are tractable but require some efforts. Below are the few questions that arose in my reading of the current draft and that I would encourage you to consider in further improving this manuscript.

 Comments:

  1. In the dependent variable section (line 101-103), a category for mental health outcome is missing.
  2. The statistical analysis section could have been much clearer if the details of modelling process was included. I was struggling to find the actual difference and the background of all the models (1-3) the authors have presented. It was therefore quite hard for me to take the actual message from the final tables too. This is something that needs some more work.
  3. Again, in the same issue, what was the difference between the three models, what was the explaining power gained or lost after switching to the other model, this has to be presented. Though their work is not for different groups like in this study, Chen (2015) have effectively explained the whole process in their paper, maybe this is something to have a look at. They have used an earlier round of CGSS as well. Another work that could be helpful is that of Chen (2017), which uses data from China’s labor-force dynamic survey.

Chen, H. & Meng, T. 2015. Bonding, Bridging, and Linking Social Capital and Self-Rated Health among Chinese Adults: Use of the Anchoring Vignettes Technique. PLoS ONE [Electronic Resource], 10, e0142300.

Chen, H., Liu, Y., Li, Z. & Xue, D. 2017a. Urbanization, economic development and health: evidence from China's labor-force dynamic survey. Int J Equity Health, 16, 207.

  1. The R-squared values of each model ranged only between 0.01 to 0.08, which can be considered low but still can be argued as relevant. This must be highlighted in the limitation section.

Reviewer 3 Report

I am bit confused regarding the results obtained by authors. My main concern is related with the significance of the model. Despite authors include both in the abstract and conclusions sections some significant relations between health and the variables involved in the analysis, the R2 obtained in all models is actually very poor: 8% in the best case. I thinks there are several statistical pitfalls that should be addressed by authors before going further.

  1. Logistic regression. Authors show the R2 in each regression, but there is no information regarding the confusion matrix. This way, we cannot know how accurate is the model.
  2. Multicollinearity. No information is given regarding the correlation between variables. We need to measure whether there is or not multicollinearity.
  3. I guess that both dependant variables could be correlated. An additional analysis should carry out a logistic regression with one dependent variable, and the other one being included as an independent variable, so considering one more variable in the results reported in tables.
  4. Most variables are not significant. Authors should carry out a stepwise regression to obtain a model with just significant variables. This way, maybe some variables that are reported as significant become non significant.
  5. Tables 3-6. We see some differences in coefficients. You should try to statistically test whether these coefficients are different or not.

Reviewer 4 Report

Comments

The right to housing experienced by the UN is measured by the ptactic of the virtue of the truth of the values that it makes available to governments.

The heterogeneity of the different age groups and their regional distribution is an important, or even essential, requirement for better mental health. socially, it all starts with housing.

Suggestions

64 - There is one more "the". Delete one of them.

273 - "..."are" significantly related to health. The sentence needs "are".

279 - low-and-middle-income families; call "low" to the hyphen.

Future research

The Sub-Sample that divides the three age groups (mainly the "living space") is a salient and appropriate attempt at approach, specially in a so important country as China because of its demographic overhang.
However, the legal system should be mentioned, as well as the mention of authors of bibliographic reference.

Round 2

Reviewer 3 Report

I am very skeptical with this paper. Results are extremely poor. Authors state that "R-squared values were relatively low in our model", but they are extremely low indeed. Results are not statistically significant, so the research has nothing new to show. Modifications made by authors are very minor, while the main concerns regarding the research have not been properly addressed. My recommendation is to reject the paper.

Author Response

We very much appreciate the comments received cin the review. All changes made to the text are marked and highlighted in yellow so that you may be easily identified. More details could be seen in the attachment.
